# Analysis of Epidemiological and Evolutionary Characteristics of Seasonal Influenza Viruses in Shenzhen City from 2018 to 2024

**DOI:** 10.3390/v17060798

**Published:** 2025-05-30

**Authors:** Weiyu Peng, Hui Liu, Xin Wang, Chao Li, Shunwu Huang, Shiyu Qi, Zhongnan Hu, Xiaoying Xu, Haihai Jiang, Jinyu Duan, Hui Chen, Manyu Huang, Ying Sun, Weihua Wu, Min Jiang, Xuan Zou, Shisong Fang

**Affiliations:** 1Shenzhen Center for Disease Control and Prevention, Shenzhen 518000, China; pengweiyucas@163.com (W.P.); zzlh512@163.com (H.L.); szwxin@163.com (X.W.); lichao-365@163.com (C.L.); 19979237567@163.com (S.H.); qqishiyu@163.com (S.Q.); hu13133145307@163.com (Z.H.); duanjinyu0128@163.com (J.D.); 18738697680@163.com (H.C.); 15977146984@163.com (M.H.); sunying0804@sina.com (Y.S.); wuweihua_1984@163.com (W.W.); jiangmin_115@163.com (M.J.); 2College of Public Health, University of South China, Hengyang 421000, China; 3Public Health College, Guangdong Pharmaceutical University, Guangzhou 510000, China; 4College of Life Sciences and Medicine, Zhejiang Sci-Tech University, Hangzhou 310000, China; 5College of Basic Medical Sciences, Zhejiang Chinese Medical University, Hangzhou 310000, China; 20181005@zcmu.edu.cn; 6School of Basic Medical Sciences, Jiangxi Medical College, Nanchang University, Nanchang 330031, China; haihaijiang2020@ncu.edu.cn

**Keywords:** seasonal influenza virus, epidemiology, phylogenic analysis, antigenic drift

## Abstract

The SARS-CoV-2 pandemic and the implementation of associated non-pharmaceutical interventions (NPIs) profoundly altered the epidemiology of seasonal influenza viruses. To investigate these changes, we analyzed influenza-like illness samples in Shenzhen, China, across six influenza seasons spanning 2018 to 2024. Influenza activity declined markedly during the SARS-CoV-2 pandemic compared with the pre-pandemic period but returned to or even exceeded pre-pandemic levels in the post-pandemic era. Phylogenetic analysis of hemagglutinin (HA) and neuraminidase (NA) genes from 58 H1N1pdm09, 78 H3N2, and 97 B/Victoria isolates revealed substantial genetic divergence from the WHO-recommended vaccine strains. Notably, key mutations in the HA genes of H1N1pdm09, H3N2, and B/Victoria viruses were concentrated in the receptor-binding site (RBS) and adjacent antigenic sites. Hemagglutination inhibition (HI) assays demonstrated that most circulating viruses remained antigenically matched to their corresponding vaccine strains. However, significant antigenic drift was observed in H3N2 clade 3C.2a1b.1b viruses during the 2018–2019 season and in B/Victoria clade V1A.3a.2 viruses during the 2023–2024 season. These findings highlight the impact of NPIs and pandemic-related disruptions on influenza virus circulation and evolution, providing critical insights for future surveillance and public health preparedness.

## 1. Introduction

Seasonal influenza is a highly contagious respiratory illness that imposes a considerable global health burden each year, with an estimated 5 million cases, according to the World Health Organization (WHO) [1,2]. The emergence of the SARS-CoV-2 pandemic markedly altered the epidemiological and evolutionary characteristics of seasonal influenza, largely due to the widespread implementation of non-pharmaceutical interventions (NPIs), such as masking and social distancing [3]. During the 2020–2021 influenza season, global influenza activity reached historically low levels [4,5]. However, as the SARS-CoV-2 transmission declined and pandemic-related restrictions were gradually lifted, influenza virus circulation rebounded substantially [6]. Notably, after China’s dynamic zero-COVID policy ended in late 2022 [7], a wave of H1N1pdm09 virus epidemics emerged in early 2023 [8,9]. These developments underscore the need for a comprehensive analysis of the SARS-CoV-2 pandemic’s impact on the prevalence, transmission, and evolutionary trends of seasonal influenza viruses.

Seasonal influenza viruses include H1N1pdm09, H3N2, B/Victoria, and B/Yamagta. Among these, H1N1pdm09 and H3N2 are currently the dominant human influenza A virus subtypes responsible for annual epidemics [10]. Although seasonal influenza vaccination remains the primary countermeasure to mitigate the impact of infection, the effectiveness of the seasonal influenza vaccine ranges from only 20 to 60% [11], largely due to the induction of narrow, strain-specific immune responses. Studies have shown that HA antigenic drift (viral genome point mutations) is the primary reason for the limited effectiveness of the seasonal influenza vaccine [12]. Due to antigenic drift, currently circulating viruses can evolve to evade pre-existing neutralizing antibodies induced by prior infection, vaccination, or both.

Seasonal influenza vaccines primarily elicit antibodies against the surface glycoprotein hemagglutinin (HA), which can block viral attachment to host receptors and/or viral membrane fusion with host cells [13]. Anti-HA antibodies, particularly those targeting its immunodominant globular head domain, are generally strain-specific. Nonetheless, a subset of broadly reactive antibodies has been identified that can recognize multiple subtypes within either influenza A viruses (IAVs) or influenza B viruses (IBVs) [14,15]. Particularly, antibodies binding the less-dominant HA stalk have demonstrated remarkable cross-reactivity within IAVs [16,17] or IBVs [18], and even across both IAV and IBV lineages [19]. In addition to HA, vaccines also trigger antibodies targeting neuraminidase (NA), the second major surface glycoprotein, which cleaves sialic acid on the host cell surface, allowing progeny viruses to be released from infected cells [20]. Studies have suggested that anti-NA antibodies independently correlate with protection from infection and provide broad protection against heterologous viruses [21,22,23], even exhibiting cross-reactivity against both IAVs and IBVs by directly binding to conserved catalytic sites [24,25]. Given that antibodies to HA and NA are the major mediators of protection against infection with influenza viruses, it is essential to constantly monitor the genetic and mutational changes in the HA and NA of seasonal influenza viruses.

In this context, we collected a total of 41,428 throat swab samples and sequenced the HA and NA sequences of 58 influenza H1N1pdm09, 78 influenza H3N2, and 97 B/Victoria viruses from isolated samples. This study aimed to investigate the epidemiology, evolutionary process, and antigenic variation of seasonal influenza viruses in Shenzhen from 2018 to 2024.

## 2. Materials and Methods

### 2.1. Sample Collection and Virus Isolation

From October 2018 to August 2024, a total of 41,428 throat swabs were collected from sentinel hospitals in Shenzhen city. The swabs were placed into viral transport media, stored in a handheld portable 4C refrigerator, and transported to the laboratory within 24 h. PCR-positive samples for influenza viruses were isolated and cultured in Madin-Darby canine kidney (MDCK) cells. A 100 µl sample of a preservation solution was inoculated into an 80% to 90% confluent monolayer of MDCK cells, and 300 µl of a virus infection medium containing 2 µg/ml of TCPK–trypsin was added for MDCK cells and virus culture. The infected MDCK cells were observed daily for cytopathogenic effect (CPE). Cell cultures that presented CPE were collected and passaged in MDCK cells for a hemagglutination inhibition assay.

### 2.2. Viral RNA Extraction and HA/NA Gene Sequencing Analysis

Viral RNA was extracted from the viral cell culture supernatant using a High Pure Viral RNA Kit (Roche, Switzerland), according to the manufacturer’s instructions. RT-PCR was performed following the method described in the WHO manual. The primers used in these studies to identify HA and NA subtypes by RT-PCR were designed, and the sequences are listed in Appendix A. Circulating influenza strains were detected, and the sequences were submitted to the Global Initiative on Sharing All Influenza Data (GISAID) database (http://www.gisaid.org, accessed on 12 February 2025). The corresponding GISASID ID numbers are shown in Appendix A.

### 2.3. Phylogenetic Analysis

For the phylogenetic analysis, the methods of sample processing and sequencing analysis were based on previous studies [26]. In brief, the HA and NA genome segment sequences of WHO-recommended vaccine strains during 2019–2024 were downloaded from the GISAID database. For each of these strains, multiple sequence alignment was performed using MAFFT v7 [27]. Phylogenetic analyses of the aligned HA and NA were performed using RAxML (version 8.1.6) [28], with GTRGAMMA applied as the nucleotide substitution model and 1000 bootstrap replicates. Phylogenetic trees were visualized using FigTree (version 1.4.3).

### 2.4. Amino Acid Substitution Analysis of Antigenic Sites and Receptor-Binding Site (RBS)

The amino acid substitutions in antigenic sites and RBS were mapped onto the influenza HA protein crystal structures (PDB: 3LZG of H1N1pdm09 HA [29]; PDB: 8FAW of H3N2 HA [30]; and PDB: 2RFU of B/Victoria HA [31]). The HA antigenic sites and RBS, as defined in a previous paper [32], were overlaid onto the protein structures. Structural models were visualized and manipulated using PyMOL.

### 2.5. Hemagglutination (HA) Assay and Hemagglutination Inhibition (HI) Antibody Examination

Stock viruses were titrated by HA assay with 1% chicken red blood cells (RBCs) or guinea pig RBCs, according to previously described methods [33]. Briefly, the viral isolates were two-fold serially diluted with PBS buffer and incubated with 1% chicken or guinea pig RBCs for 30 or 60 min at room temperature. The HA titers were measured as the highest dilution that was completely hemagglutinated.

For the HI test, the sheep serum obtained from sheep immunized with the corresponding annual cell-based vaccine strains, which were provided by the Chinese National Influenza Center as part of the influenza virus identification kit (HI method), was two-fold serially diluted with PBS buffer. Following dilution, the vaccine strains or circulating strains (4 HAU/well) were added and incubated at room temperature for 30 min. Subsequently, 1% chicken or guinea pig RBCs were introduced, and the samples were incubated for an additional 30 or 60 min at room temperature. The HI titers were recorded as the highest serum dilution that completely inhibited hemagglutination.

## 3. Results

### 3.1. Epidemiological Surveillance of Seasonal Influenza Virus

To monitor the seasonal influenza virus epidemic in Shenzhen, 41,428 throat swab samples were collected from individuals with influenza-like illness (ILI) at sentinel hospitals between October 2018 and August 2024. These samples were subsequently tested for influenza viruses in the laboratory. Among these, 10,071 (24.3%) specimens tested positive, including 7190 (71.4%) for influenza A and 2881 (28.6%) for influenza B viruses. Before the SARS-CoV-2 pandemic, the positivity rate during the 2018–2019 season was 29.9%. However, during the SARS-CoV-2 pandemic, a remarkable decline in positivity was observed. In the 2019–2020 season, the positivity rate decreased to 9.0%, and it further decreased to 1.3% in the 2020–2021 season, likely due to the implementation of non-pharmaceutical interventions (NPIs) and the emergence of the SARS-CoV-2 pandemic. Then, the positivity rates were slightly higher in the monitoring years of 2021–2022, with 16.0%. Notably, the seasonal influenza positivity rates of the 2022–2023 and 2023–2024 seasons returned to or exceeded pre-pandemic levels, at 28.1% and 33.0%, respectively (Table 1). These findings suggest that the SARS-CoV-2 pandemic and associated NPIs significantly affected the prevalence and transmission dynamics of seasonal influenza viruses in Shenzhen.

An analysis of the circulating influenza virus strains revealed notable temporal variations. During the 2018–2019 season, the H1N1pdm09 and B/Victoria viruses were predominant. Seasonal H3N2 emerged as the predominant strain in the 2019–2020 season. The 2020–2021 season saw a drastic reduction in influenza cases, with minimal viral detection. In the 2021–2022 season, the H3N2 and B/Victoria viruses were the most frequently detected strains. The 2022–2023 season was dominated by H1N1pdm09, while the 2023–2024 season exhibited a predominance of seasonal H3N2, followed by B/Victoria and H1N1pdm09. Notably, B/Yamagata viruses were absent from circulation after the 2019–2020 season (Figure 1). These results indicate that the SARS-CoV-2 pandemic may have influenced the circulating strains and lineages of seasonal influenza viruses.

Seasonal influenza activity exhibited significant variations in intensity and timing across the study period. During the 2018–2019 season, the influenza activity was moderate, with a typical rise beginning in November and sustained elevation from mid-December to June. The 2019–2020 season showed a sharp decline in activity by late March. The 2020–2021 season was marked by historically low influenza activity. In the 2021–2022 season, two distinct waves of activity occurred: B/Victoria viruses dominated from late July 2021 to April 2022, and H3N2 viruses circulated predominantly from May 2022 to August 2022. During the 2022–2023 season, the influenza activity rebounded to pre-pandemic levels. The 2023–2024 season was characterized by three consecutive waves, with the longest circulation period occurring from June 2023 to August 2024, and the highest positivity rates of the entire 2018–2024 period (Figure 1). These results demonstrate that the influenza activity and circulating characteristics of seasonal influenza viruses in Shenzhen significantly differed before, during, and after the SARS-CoV-2 pandemic.

### 3.2. Phylogenetic Analysis of Seasonal Influenza Virus

To better understand the evolutionary patterns of the seasonal influenza strains in Shenzhen during the 2018–2024 seasons, phylogenetic analyses of the HA and NA gene sequences were conducted, comparing local strains with global circulating strains, representative clades, and vaccine strains. The phylogenetic analysis of 58 H1N1pdm09 HA gene sequences revealed that circulating strains were predominantly divergent from recommended vaccine strains (Figure 2A; GISAID accession numbers in Appendix A). During the 2018–2019 season, most isolates (53.3%) clustered with clade 6B.1A.5a, while others were distributed among clades 6B.1A, 6B.1A.7, and 6B.1A.5a.1. None of the isolates matched the vaccine strain A/Michigan/45/2015 (clade 6B.1). Similarly, in the 2022–2023 season, most strains clustered within clade 6B.1A.5a.2a, with one strain (A/Shenzhen/15/2023) in clade 6B.1A.5a.2a.1. These isolates did not align with the vaccine strain A/Victoria/2570/2019 (clade 6B.1A.5a.2). By 2023–2024, all strains fell within clade 6B.1A.5a.2a.1, diverging from the vaccine strain A/Victoria/4897/2022. The NA phylogeny yielded highly consistent results, and the major circulating strains did not match the vaccine strains (Figure 2B). Therefore, these results indicate a mismatch between the circulating H1N1pdm09 viruses in Shenzhen and the recommended vaccine strains during the 2019–2024 period.

A phylogenetic analysis of 78 HA genes of influenza virus H3N2 from 2018 to 2024 was conducted (Figure 3A; GISAID accession numbers are listed in Appendix A). Additional reference sequences corresponding to the vaccine strains were included in the analysis. During the 2018–2019 season, circulating H3N2 strains were categorized into two evolutionary branches. Among these, 63.6% (n = 7) of the strains clustered within clade 3C.2a1b.1b, while 36.4% (n = 4) were grouped into clade 3C.2a1b.2. In contrast, the vaccine strain A/Singapore/INFIMH-16-0019/2016 was classified in clade 3C.2a1. For the 2021–2022 season, all circulating strains were found within clade 3C.2a1b.2a.1a.1, which closely matched the vaccine strain A/Cambodia/E0826360/2020 located in clade 3C.2a1b.2a.1a. During the 2022–2023 season, most circulating trains belonged to clade 3C.2a1b.2a.2a.3a.1, which mismatched the vaccine strain A/Darwin/9/2021, which belonged to clade 3C.2a1b.2a.2a. In contrast, in the 2023–2024 season, 100.0% (n = 20) of circulating strains exhibited a high degree of similarity to the vaccine strain A/Thailand/8/2022, which belonged to clade 3C.2a1b.2a.2a.1b. A similar trend was observed in the phylogenetic analysis of the neuraminidase (NA) gene, which yielded highly consistent results (Figure 3B). These findings indicate that circulating H3N2 strains in Shenzhen were largely divergent from vaccine strains during most influenza seasons, except the 2023–2024 season.

A phylogenetic analysis of the HA gene sequences was carried out for 97 representative influenza B/Victoria isolates from 2019 to 2024 (Figure 4A; GISAID accession numbers are listed in Appendix A). The circulating B/Victoria viruses were compared with the reference vaccine strains (B/Colorado/06/2017, B/Washington/02/2019, and B/Austria/1359417/2021). During the 2018–2019 season, 94.1% of circulating strains clustered in clade V1A.3, which differed from the vaccine strain B/Colorado/06/2017 in clade V1A.1. For the 2020–2021 season, circulating strains of B/Victoria were divided into two evolutionary branches: V1A.3a.1 and V1A.3a.2. Among these, 80.8% (n = 21) of isolates were classified into clade V1A.3a.2, while 19.2% (n = 5) belonged to clade V1A.3a.1. Similarly, clade V1A.3a.2 remained predominant, accounting for 91.7% of circulating strains during the 2021–2022 season. During this period, the vaccine strain B/Washington/02/2019, classified in clade V1A.3, showed a mismatch with circulating strains. However, all circulating B/Victoria strains during the 2022–2024 seasons fell within clade V1A.3a.2, aligning with the recommended vaccine strain B/Austria/1359417/2021. The NA gene evolution of circulating B/Victoria viruses mirrored that of the HA gene, with a mismatch observed for the 2019–2022 seasons but a match with the vaccine strain during the 2022–2024 period.

### 3.3. Amino Acid Variations of Seasonal Influenza Viruses

As we all know, antigenic drift in the HA protein is a major contributor to the limited effectiveness and narrow breadth of current influenza vaccines [34]. To better understand this phenomenon, we analyzed amino acid substitutions in the HA protein of H1N1pdm09, H3N2, and B/Victoria influenza viruses isolated between 2019 and 2024, using the WHO-recommended cell-based vaccine strains as reference controls (https://www.who.int/teams/global-influenza-programme/vaccines/who-recommendations, accessed on 12 February 2025). For the H1N1pdm09 virus, we identified a total of 12 amino acid substitutions within the antigenic sites and receptor-binding site (RBS) compared with the corresponding vaccine strain. In the 2018–2019 season, the following substitutions were observed: S77R in the Cb epitope, S167T in the Sa epitope, and T188I, A189T, and Q192E in the Sb epitope. Additionally, three substitutions were observed in RBS-N132D, S186P, and D190A. During the 2022–2023 season, the K145R substitution emerged in the antigenic site Ca2, while A189T and Q192E persisted in the antigenic site Sb. A novel substitution, E227A, was also identified in the RBS. By the 2023–2024 season, further antigenic changes included R145K in the Ca2 site, K172Q in the Ca1 site, and S193T in the RBS. These findings highlight a concentration of amino acid substitutions within the RBS and antigenic site Sb (Figure 5A), underscoring their potential role in driving antigenic drift. Importantly, no neuraminidase inhibitor (NAI) resistance-associated substitutions were detected in the neuraminidase (NA) gene of the circulating H1N1pdm09 viruses.

Compared with the WHO-recommended cell-based vaccine strain A/Singapore/INFIMH-16-0019/2016, H3N2 viruses circulating during the 2018–2019 influenza season exhibited multiple amino acid substitutions in the HA antigenic sites. These included A138S, T128A, K160T/I, F193S, S198P, S219F, T131K, T135K, S137F, P194L, and G225D. In the 2021–2022 season, additional changes were observed. Specifically, antigenic site B accumulated substitutions such as N158K and variants at position 160 (T160K/A/R/I), while antigenic site C showed the I48T substitution. Further comparison with the more recent vaccine strain A/Darwin/6/2021 revealed continued antigenic evolution. Viruses belonging to clade 3C.2a1b.2a.2a.1b harbored the I140K substitution in antigenic site A. In contrast, clade 3C.2a1b.2a.2a.3a.1 viruses exhibited both N122D and I140K substitutions in site A, along with E50K and G53N substitutions in site C. These results demonstrate that major substitutions occurred predominantly in the RBS and antigenic site B, suggesting ongoing antigenic variation in circulating H3N2 viruses (Figure 5B). Notably, as observed in H1N1pdm09, no NAI resistance substitutions were detected in the NA gene of these H3N2 viruses.

We next analyzed circulating B/Victoria-lineage viruses in comparison with the WHO-recommended cell-based vaccine strain. During the 2018–2019 influenza season, viruses belonging to clade V1A.3 exhibited G129D and G133R substitutions in the antigenic site A, along with K136E in the RBS. In the 2020–2022 seasons, viruses from clade V1A.3a.1 showed R133G in the site A, N150K in the site B, and N194D in the RBS. Concurrently, clade V1A.3a.2 viruses displayed additional substitutions, including H122Q and A127T in site A; P144L in site B; and T196N and K200R in the RBS. From 2022 to 2024, further evolution was observed within clade V1A.3a.2, with newly emerged substitutions, such as E128G and D129N in antigenic site A and D194E in the RBS. Collectively, these findings indicate that most amino acid substitutions in the HA of circulating B/Victoria viruses were concentrated in the RBS and antigenic site A, reflecting ongoing antigenic drift (Figure 5C). Consistent with our observations in H1N1pdm09 and H3N2 viruses, no NAI-resistance substitutions were detected in the NA protein.

### 3.4. Antigenic Characterization of Circulating Influenza Viruses

To assess whether genetic changes in the HA of circulating influenza viruses impacted antigenicity, HI assays were conducted using post-infection sheep antisera raised against reference cell-based vaccine strains. These assays covered H1N1pdm09-, H3N2-, and B/Victoria-lineage viruses. A total of 43 influenza A virus isolates and 16 B/Victoria isolates were obtained from patient swab samples and subsequently propagated in MDCK cells. All isolates achieved HA titers of at least 4 HAU/25 μl, ensuring their suitability for antigenic analysis. For H1N1pdm09 viruses, the HI titers ranged from 1:320 to 1:2560, with less than a four-fold reduction compared with the homologous vaccine strain, indicating preserved antigenic similarity. Among the H3N2 viruses, strains from the 2018–2019 season, belonging to clade 3C.2a1b.1b, exhibited HI titers between 1:80 and 1:320, corresponding to a four- to eight-fold reduction when compared with the A/Singapore/INFIMH-16-0019/2016 vaccine strain (clade 3C.2a1), suggesting notable antigenic drift. In contrast, clade 3C.2a1b.2 viruses retained effective reactivity with vaccine-induced antibodies. In the 2021–2022 season, clade 3C.2a1b.2a.1a.1 strains exhibited HI titers of 1:160 to 1:320, with a two- to four-fold reduction relative to the corresponding vaccine strain, suggesting limited antigenic drift. Furthermore, during the 2022–2023 and 2023–2024 seasons, circulating H3N2 strains showed no significant differences in HI titers relative to the WHO-recommended vaccine strains. For B/Victoria-lineage viruses, no significant antigenic difference was observed during the 2018–2019 and 2022–2023 seasons. However, clade V1A.3a.2 viruses circulating during the 2023–2024 season exhibited a two- to four-fold reduction in HI titers compared with the vaccine strain, indicating emerging antigenic divergence (Appendix A).

## 4. Discussion

This study provides critical insights into the epidemiology, genetic evolution, and antigenic drift of seasonal influenza viruses circulating in Shenzhen over a six-year period (2018–2024), with a particular focus on the impact of the SARS-CoV-2 pandemic. The findings underscore the dynamic nature of influenza virus circulation and highlight the profound influence of NPIs on influenza activity.

Influenza transmission in China exhibits distinct regional differences. In northern regions, influenza activity typically peaks during the winter months, especially from December to March [9,10,35]. In contrast, our surveillance data from Shenzhen, a representative city in southern China, demonstrated a broader seasonal window, extending from November to the early summer months (June–July), both before and after the SARS-CoV-2 pandemic. During the pandemic, influenza activity declined sharply, with the positivity rate dropping from 29.9% in the 2018–2019 season to just 1.3% in 2020–2021. This dramatic reduction mirrored global trends observed across Europe, the Americas, Africa, and other regions of China [4,36,37], largely attributable to widespread NPIs, such as mask wearing, social distancing, travel restrictions, and school closures [38,39]. Following the relaxation of NPIs and the resumption of population mobility, influenza activity rebounded markedly during the 2022–2024 seasons, with positivity rates reaching 28.1% in 2022–2023 and 33.0% in 2023–2024. These findings suggest that the relaxation of NPIs and increased population mobility facilitated the resurgence of influenza transmission [40]. The temporal shift in dominant subtypes, with H1N1pdm09 predominating in the 2022–2023 season, followed by H3N2 in the 2023–2024 season, aligns with global reports of post-pandemic strain replacement and indicates a reshaping of influenza virus circulation patterns [10,41]. Importantly, the continued absence of B/Yamagata-lineage viruses since the 2019–2020 season is consistent with global trends and raises the possibility of lineage extinction, with important implications for future vaccine strain selection [5,42].

Antigenic drift remains a major challenge for influenza control. Our phylogenetic analysis revealed that circulating H1N1pdm09 strains predominantly belonged to genetic clades distinct from the WHO-recommended vaccine strains. Similarly, H3N2 viruses exhibited substantial divergence in most seasons, with the exception of 2023–2024, when circulating strains closely matched with the vaccine strain A/Thailand/8/2022. B/Victoria-lineage viruses exhibited persistent mismatch with vaccine strains from 2018 to 2022, followed by closer genetic alignment with B/Austria/1359417/2021 during the 2022–2024 period. Notably, incongruities between HA and NA phylogenies were observed in both H1N1 and B/Victoria viruses. For example, the HA gene of the 2022–2023 H1N1 vaccine strain was positioned in the middle of the HA phylogenetic tree, whereas its NA gene clustered with older strains from the 2018–2019 season, suggesting a possible reassortment or differential selective pressure on these segments. A similar pattern was observed in B/Victoria viruses from the 2021–2022 season, in which most HA genes (clade V1A.3a.2) clustered in the upper portion of the phylogeny, while NA genes were located in more basal lineages. These topological discrepancies imply that HA and NA genes may be subject to different evolutionary dynamics, possibly driven by immune pressure or segment reassortment. Together, these findings are consistent with previous reports showing that genetic mismatch between vaccine and circulating strains can substantially impair vaccine effectiveness [43,44] and highlight the importance of dual-segment (HA and NA) monitoring in future vaccine strain selection efforts [8,45,46,47].

Detailed amino acid analysis further highlighted extensive antigenic drift. In H1N1pdm09 viruses, we identified 12 key substitutions in HA antigenic sites and the RBS. During the 2018–2019 season, N132D, S186P, and D90A substitutions in the HA RBS may have influenced the antigenic properties of H1N1pdm09 viruses. Previous studies have reported that the S186P change may increase the receptor-binding affinity of HA by changing the hydrogen bond network within the 190-helix, thereby enhancing the virus’s ability to infect [48]. Similarly, H3N2 viruses harbored notable 18 substitutions in the HA antigenic sites and the HA RBS. Previous studies reported that the A/Delaware/16/18 strain retains the N158 glycosylation site, and the F193S substitution can escape the neutralization ability of the A2.91.3 antibody [49]. In addition, the F193S substitution can cause the virus to have a reduced ability to neutralize the antiserum against the A/Hong Kong/4801/2014-like strain [50]. T131K and T135Ksubstitutions in the HA RBS, changing the antigenicity of HA, reduce the neutralizing ability of vaccine-induced antibodies against the virus, thus reducing the protective effect of the vaccine [51]. And B/Victoria-lineage viruses had seven substitutions in antigenic site A (V117I, H122Q, A127T, E128G, D129N, A130/V/T, and G133R), two substitutions in antigenic site B (P144L and N150K), and four substitutions in the RBS (K136E, N194D, T196N, and K200R). The presence of these substitutions within HA antigenic sites, alongside evident genetic and phylogenetic divergence, underscores a potential antigenic mismatch between the newly identified viral strains and both the current and forthcoming vaccine formulations [52,53]. Importantly, the absence of NAI resistance substitutions in H1N1pdm09, H3N2, and B/Victoria viruses suggests that antiviral treatments targeting NA remain effective against these circulating strains [54].

There is a significant positive correlation between HI titers and clinical protection against influenza [55,56]. The HI assay results underscore the dynamic antigenic evolution of seasonal influenza viruses and its implication for vaccine effectiveness. While the H1N1pdm09 and B/Victoria viruses largely remained antigenically stable across the study period, the emergence of drift variants in specific H3N2 and B/Victoria clades highlights the challenges of maintaining antigenic match in seasonal vaccines. Pronounced HI titers were observed in H3N2 clade 3C.2a1b.1b strains during the 2018–2019 season, but clade 3C.2a1b.2 viruses matched with A/Singapore/INFIMH-16-0019/2016. Compared with clade 3C.2a1b.2 and the vaccine strain, a new S198P substitution in clade 3C.2a1b.1b appeared at antigenic site B, which is speculated to be a key substitution affecting antigenicity. The recent antigenic divergence detected in B/Victoria clade V1A.3a.2 in 2023–2024 warrants close surveillance, as even modest reductions in HI titers can signal the beginning of significant antigenic drift. Overall, these findings reinforce the importance of continuous virological and antigenic monitoring to guide evidence-based vaccine strain selection.

Despite its strengths, this study had several limitations. First, the reliance on sentinel hospital data may not fully capture community-level influenza activity, potentially leading to selection bias. Second, this study did not assess the impact of vaccination rates, population immunity, or co-infections with other respiratory viruses, which could influence influenza dynamics. Third, influenza virus isolation was performed using standard MDCK cells rather than MDCK-SIAT1 cells. The latter are better suited for isolating recent H3N2 viruses because they express more human-like receptors. Although this may have had a limited impact on our antigenic analysis, it should be considered when interpreting the results. Fourth, the lack of serum samples from well-defined vaccine recipients with known vaccination timelines and vaccine types limits the precision of our antigenic analysis, highlighting the need for future studies to focus more specifically on such populations to improve the reliability of HI-based conclusions.

## Figures and Tables

**Figure 1 viruses-17-00798-f001:**
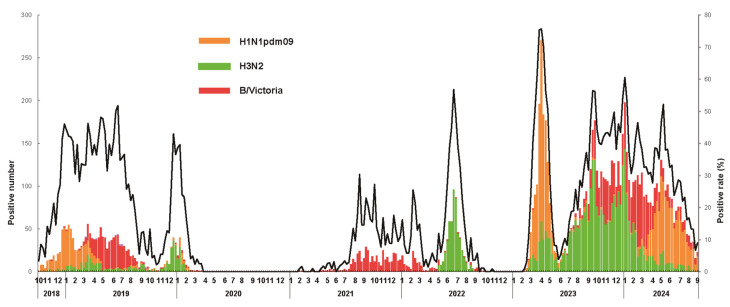
Influenza virus positivity rate among influenza-like cases in 2019–2024 in Shenzhen City. Horizontal axis: time (months); vertical axis: positivity rate. Green: H1N1pdm09 virus; orange: H3N2 influenza virus; and red: B/Victoria virus.

**Figure 2 viruses-17-00798-f002:**
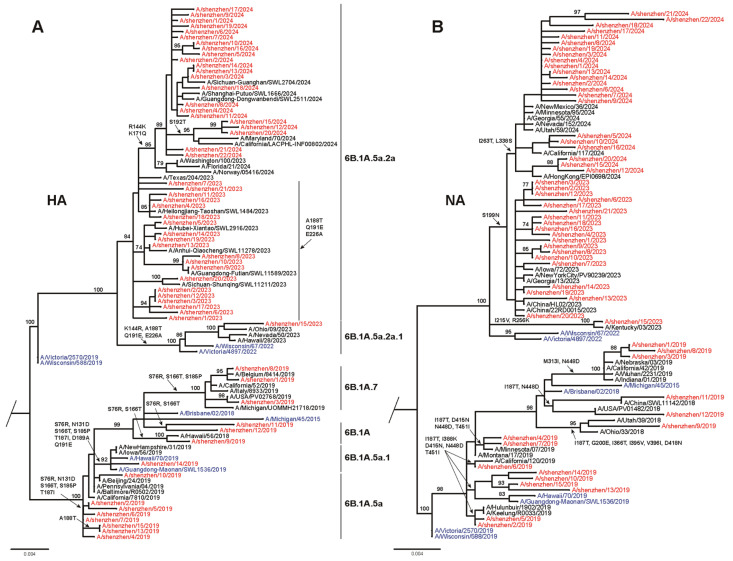
Phylogenetic analysis of nucleotide sequences of HA and NA genes of identified circulating influenza H1N1pdm09 viruses. (**A**) Phylogenetic tree of HA gene. (**B**) Phylogenetic tree of NA gene. Red indicates strains isolated in this study. The recommended vaccine strains are marked by the blue color. Compared with the vaccine strain, amino acid substitutions in the HA antigenic sites and RBS are presented (H1 number), along with substitutions in the head domain of NA (N1 number).

**Figure 3 viruses-17-00798-f003:**
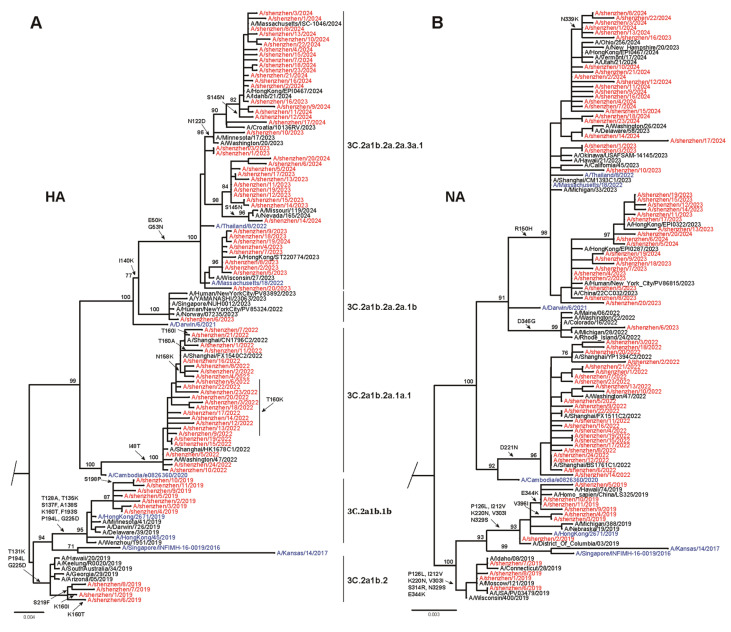
Phylogenetic analysis of nucleotide sequences of HA and NA genes of identified circulating influenza H3N2 viruses. (**A**) Phylogenetic tree of HA gene. (**B**) Phylogenetic tree of NA gene. Red indicates strains isolated in this study. Recommended vaccine strains are highlighted in blue. Compared with the vaccine strain, amino acid substitutions in the HA antigenic sites and RBS are shown (H3 number), along with substitutions in the head domain of NA (N2 number).

**Figure 4 viruses-17-00798-f004:**
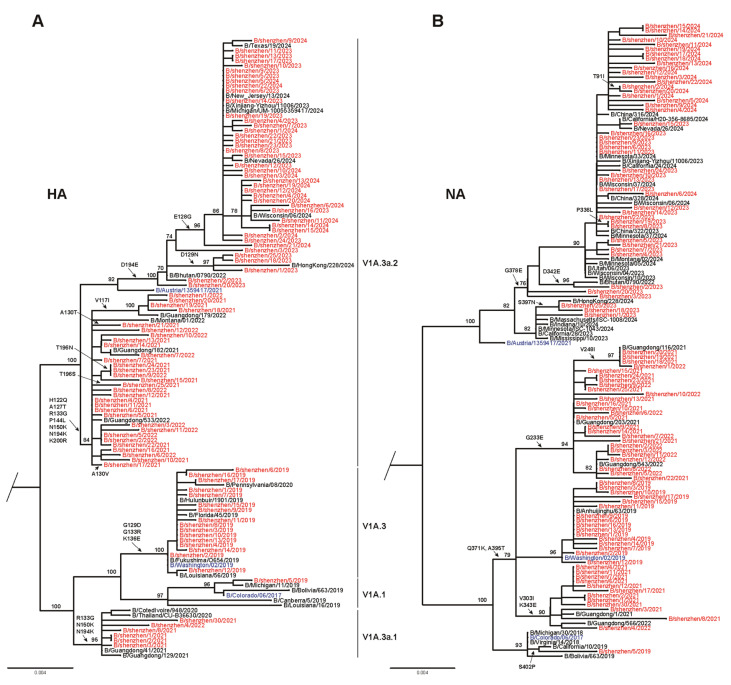
Phylogenetic analysis of nucleotide sequences of HA and NA genes of identified circulating influenza B/Victoria viruses. (**A**) Phylogenetic tree of HA gene. (**B**) Phylogenetic tree of NA gene. Red indicates strains isolated in this study. Recommended vaccine strains are highlighted in blue. Compared with the vaccine strain, amino acid substitutions in the HA antigenic sites and RBS are shown (B/Victoria HA number), along with substitutions in the head domain of NA (B/Victoria NA number).

**Figure 5 viruses-17-00798-f005:**
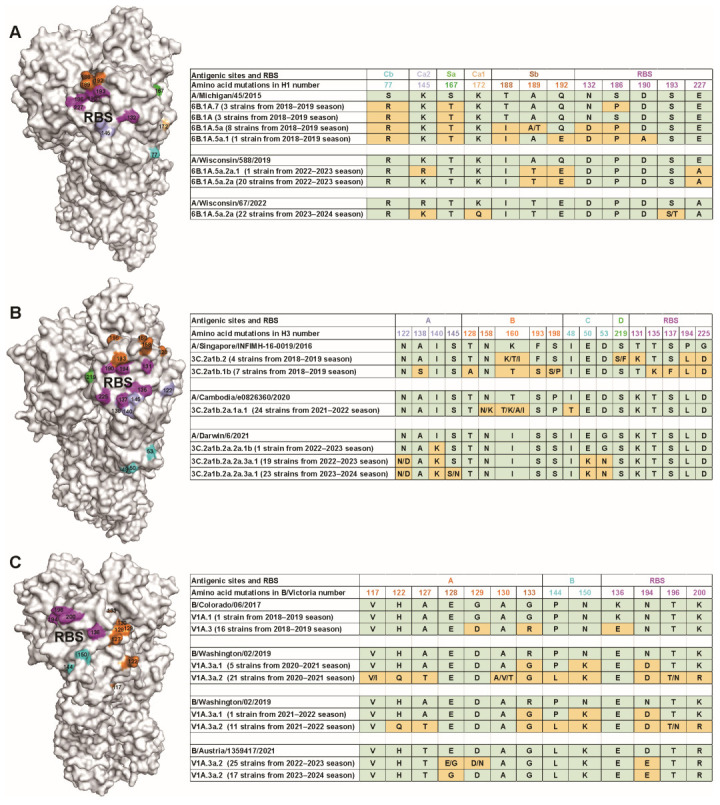
Amino acid substitutions in HA of identified viruses compared with recommended cell-based vaccine strains. (**A**) HA amino acid substitutions from A (H1N1) viruses were mapped onto crystal structure of A/California/04/2009 HA (PDB: 3LZG). (**B**) Amino acid substitutions of HA from H3N2 viruses were mapped onto crystal structure of A/Victoria/22/2020 HA (PDB: 8FAW). (**C**) HA amino acid substitutions from circulating B/Victoria strains were mapped onto crystal structure of B/Hong Kong/8/1973 HA (PDB: 2RFU). Residue substitutions are highlighted in orange.

**Table 1 viruses-17-00798-t001:** Prevalence of influenza viruses in Shenzhen city, China, during 2018–2024 seasons.

Season	Sample Number	All Influenza Virus	H1N1pdm09	H3N2	B/Victoria	B/Yamagata
		Positive Number (Positive Rate %)	Positive Number (Positive Rate %)	Positive Number (Positive Rate %)	Positive Number (Positive Rate %)	Positive Number (Positive Rate %)
2018.10–2019.09	4947	1481 (29.9%)	604 (12.2%)	258 (5.2%)	610 (12.3%)	9 (0.2%)
2019.10–2020.09	3081	278 (9.0%)	55 (1.8%)	203 (6.6%)	20 (0.7%)	0 (0.0%)
2020.10–2021.07	4150	53 (1.3%)	0 (0.0%)	0 (0.0%)	53 (1.3%)	0 (0.0%)
2021.08–2022.09	6653	1063 (16.0%)	0 (0.0%)	541 (8.1%)	522 (7.9%)	0 (0.0%)
2022.10–2023.05	5249	1473 (28.1%)	1082 (20.6%)	388 (7.4%)	3 (0.1%)	0 (0.0%)
2023.06–2024.08	17,348	5722 (33.0%)	1259 (7.3%)	2800 (16.1%)	1664 (9.6%)	0 (0.0%)

## Data Availability

The data that support the findings of this study are openly available in GISAID at https://gisaid.org/; the reference number is in the supporting data.

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
