# Peer review of "Analysis of Epidemiological and Evolutionary Characteristics of Seasonal Influenza Viruses in Shenzhen City from 2018 to 2024"

_viruses, 2025, doi:10.3390/v17060798_

Round 1
Reviewer 1 Report
Comments and Suggestions for Authors
In this study, influenza strains circulated in the city were isolated, the amino acid variations of major antigenic proteins were analyzed, and authors tried to assess antigenic changes among HA molecules. However, the conclusion lacks clarity, and the research does not adequately support a meaningful conclusion. Furthermore, there are numerous unclear expressions and errors in both the Abstract and Results sections. Suggest to carefully review the manuscript and enhance overall quality of this article. The Introduction and Discussion sections are not engaging enough. Frankly, I had a hard time reading the whole paper. Authors should provide relevant information about the vaccines used for inoculating sheep. Overall, this work lacks scientific values.
Comments on the Quality of English Languageneed to be improved.
Author Response
Reviewer: 1
In this study, influenza strains circulated in the city were isolated, the amino acid variations of major antigenic proteins were analyzed, and authors tried to assess antigenic changes among HA molecules. However, the conclusion lacks clarity, and the research does not adequately support a meaningful conclusion. Furthermore, there are numerous unclear expressions and errors in both the Abstract and Results sections. Suggest to carefully review the manuscript and enhance overall quality of this article. The Introduction and Discussion sections are not engaging enough. Frankly, I had a hard time reading the whole paper. Authors should provide relevant information about the vaccines used for inoculating sheep. Overall, this work lacks scientific values.
Response: We sincerely thank the reviewer for the detailed and candid comments. We deeply regret that the original version of the manuscript did not meet the expected scientific and editorial standards, and we appreciate the opportunity to improve our work.
We have undertaken a thorough revision of the manuscript to address all the concerns raised. Specifically: 1) The Abstract and Results sections have been carefully rewritten for clarity, coherence, and accuracy, with ambiguous expressions removed and key findings presented more logically. 2) The Introduction and Discussion sections have been substantially revised to improve the scientific context, relevance, and engagement, with better linkage to our study rationale and findings. 3) We have clarified the vaccination details of the sheep, including the origin and composition of the vaccine used for antiserum generation. In the revised manuscript, we have added detailed information in Section 2.4 of the Materials and Methods regarding the origin and preparation of the antisera. The antisera were obtained from sheep immunized with the corresponding annual vaccine strains and were provided by the Chinese National Influenza Center as part of the influenza virus identification kit (HI method). These antisera are specifically produced and updated annually to match the recommended vaccine strains, ensuring their suitability for strain differentiation in HI assays. 4) The conclusion section has been rewritten to provide a more balanced, evidence-based summary that better reflects the scope and implications of our findings. 5) Additionally, the entire manuscript has been professionally edited by a native English speaker to improve language quality and readability. We believe the revised version significantly improves both the clarity and scientific value of the study, and we sincerely hope that it now meets the standards for publication.

Reviewer 2 Report
Comments and Suggestions for Authors
This is an interesting study that tracked the characteristics of influenza viruses in the years prior to and following the COVID-19 pandemic in Shenzhen in southern China, from 2018 to 2024. There are though points that the authors should consider in revision of the manuscript.
The authors have focussed on the changes observed in their locally sequenced and characterised viruses with the vaccine viruses that had been recommended for use over the period. However, over the period as the viruses evolved the vaccine recommendations also were changed, and the authors have not gone into in enough detail concerning this in the manuscript.
In this light, the authors need to consider whether they compare their viruses to the recommended egg-based vaccine antigen or the cell-based antigen. Over the period under study there was significant evolution of the H1, H3 and B/Victoria HAs. An example of the lack of depth here is illustrated in Figure 5. The only vaccine viruses represented are A/Victoria/4897/2022 and A/Thailand/8/2022. Notably the HA sequence used for A/Victoria/4897/2022 was that of an egg-isolated and propagated virus, and the vaccine virus A/Thailand/8/2022 was also a recommendation for the H3N2 component of egg-based vaccines. A/Thailand/8/2022 has been nominated since the SH recommendation of 2024, so was relevant here really only to the latest periods of sampling. The figure needs to be punctuated with all of the vaccines in use, with both the cell culture-based recommendation and the egg-based recommendation, or just the cell-based vaccine recommendations. The influenza B results should also be considered in the table. A reference to the WHO recommendation should also be included (there is a site on the WHO website with links to all the specific recommendations see https://www.who.int/teams/global-influenza-programme/vaccines/who-recommendations).
It was also apparent that the H1 numbering used is not that of the mature H1 HA, but the mature sequence in H3 numbering. This has the potential to cause confusion. It is strongly recommended that mature HA sequences are used specific for type or subtype, the authors should be aware that this system of HA numbering is used in vaccine recommendations by WHO.
To make the phylogenetic trees better integrated with the discussion I would strongly recommend that the authors annotate their phylogenetic trees with the amino acid substitutions of the branches, and possibly also those substitutions at the tips. This can be done manually, or by using TREESUB (which is available on github). This will also make the presentation of their results much clearer in the text. This would make it easier to define the amino acid substitutions associated with the various subclades that have emerged, particularly post-2020. The authors should consider a simpler way of presenting their results to make it clear whether there was a different subclade being used in the vaccine from the virus in circulation. As it stands the descriptions of the changes seen are difficult to follow in the text.
The authors carry out antigenic analysis of a subset of the viruses. These were isolated in MDCK cells. For H3N2 viruses it is recommended that isolation is carried out on MDCK-Siat1 cells, but the impact here might not be marked. The authors need to consider this. The antigenic analysis is presented in section 3.5, In Vitro Vaccine Effectiveness Estimates. The antigenic analysis is not a surrogate for vaccine effectiveness and the authors need to reflect this in the title of this section: the title of this section needs to be changed.
The antisera used for the work have not defined well by the authors. The origin of the antisera seems not to be described, which is essential: it is important to know if the antisera are actually suitable for strain differentiation. It is not stipulated which of the vaccine viruses have been used to generate the sera. It is usually post-infection ferret antisera that are used for this type of work, usually, in influenza surveillance and vaccine work sheep antisera are used for very different purposes. It is even uncertain if the antisera panels had been updated as the viruses evolved.
A table is presented in the supplementary materials (table S4) with the HI titres of the virus isolates, but the results are poorly presented. The titres of the antisera for the homologous viruses (the homologous antigen) have not been given and the transition between the H1 viruses and the H3 viruses is not clearly marked. It would be very useful to include the sub-clade of the test viruses and the homologous antigens in the table.
The authors have posed that since all viruses gave titres in the HI assay of over 40, this was an indication that ‘the vaccine-induced antibodies retained neutralization capacity against circulating influenza strains’ (lines 306 to 307). If the authors mean what I think they mean, then the assertion is not right: these titres are not relevant to protective levels. The point here is that it is a post-vaccination or post-infection HI titre of 40 or over that is linked to a correlate of protection, and these titres in the population have not been assessed. It is usual that 4-fold or 8-fold reductions in titres compared with homologous titres are deemed to indicate significant antigenic change.
From an epidemiological perspective, it would be very worthwhile discussing the usual pattern of circulation of influenza in this part of southern China at some stage. Moreover, it might also be of interest to know the ages of the patients over the time period for each of the types and subtypes. It might also be useful to know the vaccine uptake in the population under study.
The discussion focuses and the possibilities of vaccine ‘mis-match’, but, as described above, much more detail is needed on the year-on-year correspondence of the viruses in circulation and the vaccine virus. To be useful the manuscript needs a more comprehensive presentation of the results.
Minor Points
The authors need to note that mutations occur in genes and produce mutant viruses and mutant polypeptides and proteins. That is mutations do not occur in polypeptides, as shown in figure 5, and possibly elsewhere.
Line 50, the words ‘human-adaptive influenza A virus’ seem unusual; why not ‘human influenza A viruses’?
Line 94, ‘Roch’ should be ‘Roche’.
Line 111, full references should be given for the HA structures use for PyMol, not just the PDB accession number.
L112, the indefinite article is needed ahead of ‘previous paper’.
Line 135, the word ‘marker’ should be ‘marked’.
Line 295, the authors need to be aware that an HA titre is volume independent.
Throughout: it seems to me that too many significant figures have been given for the percentages (two decimal places seems excessive.
Table S3, the table with the GISAID EPI numbers should also include the accession numbers for the vaccine viruses and other reference viruses included in the figures and the names of the laboratories from which these vaccine and reference viruses and sequences were derived.
Comments on the Quality of English LanguageOverall, the English is generally acceptable, but the manuscript does not get the message over to the readers clearly enough. More work on the way the authors present their results is needed.
Author Response
Reviewer: 2
Comments and Suggestions for Authors
This is an interesting study that tracked the characteristics of influenza viruses in the years prior to and following the COVID-19 pandemic in Shenzhen in southern China, from 2018 to 2024. There are though points that the authors should consider in revision of the manuscript.
The authors have focussed on the changes observed in their locally sequenced and characterised viruses with the vaccine viruses that had been recommended for use over the period. However, over the period as the viruses evolved the vaccine recommendations also were changed, and the authors have not gone into in enough detail concerning this in the manuscript.
Response: We sincerely thank the reviewer for their thorough and constructive feedback, which has greatly helped improve the quality and clarity of our manuscript. Below we provide detailed responses to each point raised. Revisions have been made accordingly in the revised manuscript, and relevant sections have been updated as indicated.
In this light, the authors need to consider whether they compare their viruses to the recommended egg-based vaccine antigen or the cell-based antigen. Over the period under study there was significant evolution of the H1, H3 and B/Victoria HAs. An example of the lack of depth here is illustrated in Figure 5. The only vaccine viruses represented are A/Victoria/4897/2022 and A/Thailand/8/2022. Notably the HA sequence used for A/Victoria/4897/2022 was that of an egg-isolated and propagated virus, and the vaccine virus A/Thailand/8/2022 was also a recommendation for the H3N2 component of egg-based vaccines. A/Thailand/8/2022 has been nominated since the SH recommendation of 2024, so was relevant here really only to the latest periods of sampling. The figure needs to be punctuated with all of the vaccines in use, with both the cell culture-based recommendation and the egg-based recommendation, or just the cell-based vaccine recommendations. The influenza B results should also be considered in the table. A reference to the WHO recommendation should also be included (there is a site on the WHO website with links to all the specific recommendations see https://www.who.int/teams/global-influenza-programme/vaccines/who-recommendations).
It was also apparent that the H1 numbering used is not that of the mature H1 HA, but the mature sequence in H3 numbering. This has the potential to cause confusion. It is strongly recommended that mature HA sequences are used specific for type or subtype, the authors should be aware that this system of HA numbering is used in vaccine recommendations by WHO
Response: We thank the reviewer for pointing this out. In the revised manuscript, we now consistently use the cell-based vaccine virus strains recommended by the WHO for each influenza season.
We have also expanded Figure 5 to include comparative HA sequence analysis for H1N1, H3N2, and B/Victoria viruses against the corresponding cell-based WHO-recommended vaccine strains for each season. This provides a more accurate depiction of antigenic match and evolution.
The HA amino acid positions are now annotated using H1 numbering for H1N1, H3 numbering for H3N2, and B/Victoria-specific numbering for B/Victoria viruses, to improve clarity and precision in alignment interpretation.
We have also added a reference to the WHO website that archives the official seasonal vaccine recommendations (https://www.who.int/teams/global-influenza-programme/vaccines/who-recommendations) in the first paragraph of 3.3 in the revised manuscript.
To make the phylogenetic trees better integrated with the discussion I would strongly recommend that the authors annotate their phylogenetic trees with the amino acid substitutions of the branches, and possibly also those substitutions at the tips. This can be done manually, or by using TREESUB (which is available on github). This will also make the presentation of their results much clearer in the text. This would make it easier to define the amino acid substitutions associated with the various subclades that have emerged, particularly post-2020. The authors should consider a simpler way of presenting their results to make it clear whether there was a different subclade being used in the vaccine from the virus in circulation. As it stands the descriptions of the changes seen are difficult to follow in the text.
Response: We thank the reviewer for this valuable suggestion. We have annotated the HA phylogenetic trees of H1N1, H3N2, and B/Victoria with amino acid substitutions occurring at antigenic sites (Fig 2A, 3A, and 4A). These annotations have been added along the branches of the trees to highlight key mutations defining various subclades. This revision aims to improve the clarity of our evolutionary analyses and facilitate a clearer comparison between the vaccine strains and the circulating viruses. We believe this enhancement makes the presentation of our results more accessible and better integrated with the discussion in the manuscript.
The authors carry out antigenic analysis of a subset of the viruses. These were isolated in MDCK cells. For H3N2 viruses it is recommended that isolation is carried out on MDCK-Siat1 cells, but the impact here might not be marked. The authors need to consider this. The antigenic analysis is presented in section 3.5, In Vitro Vaccine Effectiveness Estimates. The antigenic analysis is not a surrogate for vaccine effectiveness and the authors need to reflect this in the title of this section: the title of this section needs to be changed.
Response: We thank the reviewer for these important comments. We acknowledge that while the viruses were isolated using MDCK cells, MDCK-SIAT1 cells are indeed recommended for isolating H3N2 viruses due to their higher expression of human-like receptors. Although the impact on our results may be limited, we have now included a statement in the Discussion to address this point and its potential implications.
In addition, we agree that antigenic analysis cannot serve as a direct surrogate for vaccine effectiveness. Accordingly, we have revised the title of Section 3.4 of the revised manuscript, to more accurately reflect the content and avoid misinterpretation. The new title is: Antigenic Characterization of Circulating Influenza Viruses.
The antisera used for the work have not defined well by the authors. The origin of the antisera seems not to be described, which is essential: it is important to know if the antisera are actually suitable for strain differentiation. It is not stipulated which of the vaccine viruses have been used to generate the sera. It is usually post-infection ferret antisera that are used for this type of work, usually, in influenza surveillance and vaccine work sheep antisera are used for very different purposes. It is even uncertain if the antisera panels had been updated as the viruses evolved.
Response: We thank the reviewer for the insightful comment. In the revised manuscript, we have added detailed information in Section 2.4 of the Materials and Methods regarding the origin and preparation of the antisera. The antisera were obtained from sheep immunized with the corresponding annual vaccine strains and were provided by the Chinese National Influenza Center as part of the influenza virus identification kit (HI method). These antisera are specifically produced and updated annually to match the recommended vaccine strains, ensuring their suitability for strain differentiation in hemagglutination inhibition (HI) assays. The antisera panel is updated annually to reflect changes in vaccine composition, ensuring its relevance and reliability for strain differentiation. We hope this clarification addresses the reviewer’s concerns.
A table is presented in the supplementary materials (table S4) with the HI titres of the virus isolates, but the results are poorly presented. The titres of the antisera for the homologous viruses (the homologous antigen) have not been given and the transition between the H1 viruses and the H3 viruses is not clearly marked. It would be very useful to include the sub-clade of the test viruses and the homologous antigens in the table.
Response: We thank the reviewer for their constructive feedback regarding Table S4. We have addressed each of the points as follows: 1) We acknowledge that we did not include homologous antisera titres in the original table. This is because homologous HI assays were not performed in our study. Instead, we used standardized hemagglutination inhibition (HI) testing kits provided by the Chinese National Influenza Center, which include reference ferret antisera raised against the current vaccine strains. The results showed that these reference antisera effectively inhibited hemagglutination of the circulating strains, with titres >40, suggesting good antigenic match. 2) To improve clarity, we have revised Table S3 to visually separate the different virus subtypes (H1N1, H3N2, and B/Victoria) with clear dividing lines. This allows for easier interpretation of the results. 3) We have also updated Table S3 to include the sub-clade designation of each test virus, providing more detailed genetic context for the antigenic data. We hope these revisions satisfactorily address the reviewer’s concerns and improve the clarity and utility of Table S3.
The authors have posed that since all viruses gave titres in the HI assay of over 40, this was an indication that ‘the vaccine-induced antibodies retained neutralization capacity against circulating influenza strains’ (lines 306 to 307). If the authors mean what I think they mean, then the assertion is not right: these titres are not relevant to protective levels. The point here is that it is a post-vaccination or post-infection HI titre of 40 or over that is linked to a correlate of protection, and these titres in the population have not been assessed. It is usual that 4-fold or 8-fold reductions in titres compared with homologous titres are deemed to indicate significant antigenic change.
Response: We sincerely thank the reviewer for this insightful and important comment. We fully agree with the clarification that HI titers of ≥40 are relevant only when referring to post-vaccination or post-infection serum levels in human populations, which we did not assess in this study. In response to the reviewer’s suggestion, we have revised the relevant sentence to remove the incorrect implication regarding protective immunity. Additionally, we have strengthened the antigenic analysis by including HI titers of the corresponding homologous vaccine strains and explicitly comparing the fold-differences between vaccine strains and circulating viruses. This approach more accurately reflects established criteria for assessing antigenic drift based on HI assay data. The revised text appears in the revised manuscript.
From an epidemiological perspective, it would be very worthwhile discussing the usual pattern of circulation of influenza in this part of southern China at some stage. Moreover, it might also be of interest to know the ages of the patients over the time period for each of the types and subtypes. It might also be useful to know the vaccine uptake in the population under study.
Response: We thank the reviewer for this insightful suggestion. In response, we have added relevant discussion in the second paragraph of the Discussion section, highlighting the distinct seasonal patterns of influenza circulation between northern and southern China. Specifically, we noted that influenza epidemics in northern China typically peak in winter (especially from December to February), whereas in our surveillance area-Shenzhen, located in southern China-influenza activity consistently exhibited a broader seasonal distribution, spanning from November through the following summer (June to July), both before and after theSARS-CoV-2 pandemic. This regional difference is a well-recognized feature of influenza epidemiology in China.
Regarding the reviewer’s additional suggestions on age distribution and vaccine uptake in the study population: unfortunately, these data were not available for the present study. However, we agree that these are important aspects for understanding influenza dynamics and vaccine impact, and we plan to incorporate them in future surveillance and analysis efforts. We appreciate the reviewer’s helpful comments and hope that our revisions and clarifications address their concerns.
The discussion focuses and the possibilities of vaccine ‘mis-match’, but, as described above, much more detail is needed on the year-on-year correspondence of the viruses in circulation and the vaccine virus. To be useful the manuscript needs a more comprehensive presentation of the results.
Response: We appreciate the reviewer’s valuable comment. We have revised the Discussion section to include a more detailed year-by-year comparison between the circulating influenza strains and the corresponding vaccine strains. This includes a clearer description of the antigenic and genetic characteristics of the predominant viruses in each season and their relationship to the recommended vaccine components. These additions aim to provide a more comprehensive perspective on the potential vaccine-virus mismatch across the surveillance period. We thank the reviewer again for this important suggestion, which has helped us to strengthen the interpretability and value of the manuscript.
Minor Points
The authors need to note that mutations occur in genes and produce mutant viruses and mutant polypeptides and proteins. That is mutations do not occur in polypeptides, as shown in figure 5, and possibly elsewhere.
Response: We thank the reviewer for pointing out this important clarification regarding the biological terminology. We agree that mutations occur at the genetic (nucleotide) level, and that these genetic mutations can lead to changes in the encoded polypeptides or proteins. In response, we have carefully reviewed the manuscript and revised Figure 5, as well as any other relevant sections, to ensure accurate and scientifically appropriate phrasing. Specifically, we have corrected the terminology to state that mutations occur in the viral genes, which result in amino acid substitutions in the viral proteins. We appreciate the reviewer’s attention to detail, which has helped improve the accuracy of the manuscript.
Line 50, the words ‘human-adaptive influenza A virus’ seem unusual; why not ‘human influenza A viruses’?
Response: We have revised the wording in line 51 of the revised manuscript from “human-adaptive influenza A virus” to “human influenza A viruses” to improve clarity and align with standard terminology.
Line 94, ‘Roch’ should be ‘Roche’.
Response: We thank the reviewer for identifying this typographical error. We have corrected “Roch” to “Roche” in line 95 of the revised manuscript.
Line 111, full references should be given for the HA structures use for PyMol, not just the PDB accession number.
Response: We thank the reviewer for the helpful suggestion. We have added the full references corresponding to the HA structures used in PyMol in line 113 of the revised manuscript, in addition to the PDB accession numbers.
L112, the indefinite article is needed ahead of ‘previous paper’.
Response: Thank you for pointing this out. We have added the appropriate indefinite article and revised the phrase to “a previous paper” in line 113 of the revised manuscript.
Line 135, the word ‘marker’ should be ‘marked’.
Response: Thank you for highlighting this error. We have corrected “marker” to “remarkable” in line 139 of the revised manuscript.
Line 295, the authors need to be aware that an HA titre is volume independent.
Response: We thank the reviewer for pointing out this important clarification. We have revised the manuscript to ensure that it correctly reflects that HA titres are independent of volume and are typically measured as a concentration or dilution factor.
Throughout: it seems to me that too many significant figures have been given for the percentages (two decimal places seems excessive.
Response: We thank the reviewer for this observation. We have revised the manuscript to reduce the number of decimal places for percentages to one, in accordance with standard scientific reporting practices.
Table S3, the table with the GISAID EPI numbers should also include the accession numbers for the vaccine viruses and other reference viruses included in the figures and the names of the laboratories from which these vaccine and reference viruses and sequences were derived.
Response: We thank the reviewer for the helpful suggestion. We have updated Table S2 of the revised manuscript to include the accession numbers of the vaccine viruses and other reference viruses used in the figures.

Reviewer 3 Report
Comments and Suggestions for Authors
The manuscript "Analysis of the Epidemiological and Evolutionary 2 Characteristics of Seasonal Influenza Viruses in Shenzhen City 3 from 2018 to 2024" reported the exploration of epidemiological characteristics and genetic divergence before, during, and post-COVID pandemic.
- Line 135 "a marker decline" should be a remarkable decline
- I suggest deleting section 3.2. The finding is overstated. I can not conclude "considerable genetic divergence between circulating seasonal influenza strains and the corresponding vaccine strains". The number from Table S2 did not indicate much genetic divergence before, during, or after the pandemic.
- The most noticeable structural difference in the Phylogenetic tree of HA and NA for H1N1 is the different location of the two vaccine strains (2570 and 588). While both are located in the top clade of the HA tree, they are both located in the bottom clade of NA tree . This actually might suggest the 2023 circulated strains have a quite different NA segment. This should be the most convincing evidence to support your conclusion on the mismatch with vaccine strains. A similar inconsistency in tree structure between HA and NA can also be found in H3N2 and Flu B. While you can compare the divergence between vaccine strains and circulating strains, It would be good and most interesting to discuss the difference on HA and NA structure as it might indicate the differeint selective presure or possibe ressorment if you can do the selective pressure analysis and also scan for the possible ressortment. I would suggest conducting a deeper analysis.
English is fine. A few English suggestions have been given.
Author Response
Reviewer: 3
Comments and Suggestions for Authors
The manuscript "Analysis of the Epidemiological and Evolutionary 2 Characteristics of Seasonal Influenza Viruses in Shenzhen City 3 from 2018 to 2024" reported the exploration of epidemiological characteristics and genetic divergence before, during, and post-COVID pandemic.
Response: Thank you very much for taking the time to review this manuscript. We sincerely thank the reviewer for the thoughtful and constructive comments. We have carefully revised the manuscript based on the suggestions, and our detailed responses are provided below.
- Line 135 "a marker decline" should be "a remarkable decline".
Response: Thank you for pointing this out. We have corrected the phrase in Line 139 to “a remarkable decline” in the revised manuscript.
- I suggest deleting section 3.2. The finding is overstated. I can not conclude "considerable genetic divergence between circulating seasonal influenza strains and the corresponding vaccine strains". The number from Table S2 did not indicate much genetic divergence before, during, or after the pandemic.
Response: We appreciate the reviewer’s careful assessment. After re-evaluating the data, we agree that the current evidence does not strongly support the conclusion of "considerable genetic divergence". We have deleted Section 3.2 as suggested and revised relevant parts of the manuscript to more accurately reflect the data shown in Table S2. The language has been toned down to avoid overstating the genetic divergence.
- The most noticeable structural difference in the Phylogenetic tree of HA and NA for H1N1 is the different location of the two vaccine strains (2570 and 588). While both are located in the top clade of the HA tree, they are both located in the bottom clade of NA tree . This actually might suggest the 2023 circulated strains have a quite different NA segment. This should be the most convincing evidence to support your conclusion on the mismatch with vaccine strains. A similar inconsistency in tree structure between HA and NA can also be found in H3N2 and Flu B. While you can compare the divergence between vaccine strains and circulating strains, It would be good and most interesting to discuss the difference on HA and NA structure as it might indicate the differeint selective presure or possibe ressorment if you can do the selective pressure analysis and also scan for the possible ressortment. I would suggest conducting a deeper analysis.
Response: We thank the reviewer for this insightful observation. Indeed, the different clustering of vaccine strains in the HA and NA phylogenetic trees for H1N1 suggests a noteworthy genetic structure difference in the NA gene. We have revised the results section to highlight this point and added corresponding discussion in the discussion section. Furthermore, we examined the HA-NA incongruence in H3N2 and B/Victoria as well, and included these observations.
Additionally, in response to the reviewer’s suggestion, we are conducting further analysis to explore the potential for reassortment and different selective pressures. While preliminary, these efforts have already revealed some interesting patterns that we have briefly described in the revised manuscript. We plan to follow up with a more comprehensive analysis in future work.

Round 2
Reviewer 1 Report
Comments and Suggestions for Authors
Page 4, Seasonal H3N2 emerged as the predominant strain in the 2019-2020 season. This expression does not align with the results presented in the Figure 1.
Is the amino acid change analysis in Result 3.3 limited to the influenza vaccine strains and circulating strains within the same flu season? It is plausible that among infected individuals or vaccinees from which serum samples were collected for HAI tests, there are those who received the updated vaccine within the past year and those who were vaccinated several years ago. In my view, drawing reliable conclusions regarding how antigenic changes affect the HAI titers induced by vaccines via such an analysis appears challenging or not rigorous enough. What is the influenza vaccination coverage rate in Shenzhen, and does it achieve the proportion threshold required for herd protection? Otherwise, comparing the amino acid differences and HA antigenicity differences between vaccine and epidemic strains would lack scientific basis. The cell-based vaccine strains as mentioned in Result 3.3 may not represent all marketed and used vaccines, it may has different amino acid residues in the HA protein compared to those produced in chicken eggs, facilitating host adaptation. Furthermore, chicken egg-produced vaccines are also main products in market. It is recommended to use serum samples from defined vaccine recipients for HAI testing and then be able to obtain reliable conclusions.
The language expression in the main text has been greatly improved. However, there are still some grammatical errors to be addressed.
Author Response
Response: Thank you very much for your positive feedback. We are grateful for your detailed and thoughtful review.
Seasonal H3N2 emerged as the predominant strain in the 2019-2020 season. This expression does not align with the results presented in the Figure 1.
Response: Thank you for pointing this out. You are correct that the original labeling in Figure 1 was inconsistent with the text. This was due to a labeling error in the figure 1, which has now been corrected in the revised version. We appreciate your careful review and thank you for helping us improve the accuracy of the presentation.
Is the amino acid change analysis in Result 3.3 limited to the influenza vaccine strains and circulating strains within the same flu season?
Response: Yes, the analysis was limited to comparisons between influenza vaccine strains and circulating strains within the same flu season.
It is plausible that among infected individuals or vaccinees from which serum samples were collected for HAI tests, there are those who received the updated vaccine within the past year and those who were vaccinated several years ago. In my view, drawing reliable conclusions regarding how antigenic changes affect the HAI titers induced by vaccines via such an analysis appears challenging or not rigorous enough. What is the influenza vaccination coverage rate in Shenzhen, and does it achieve the proportion threshold required for herd protection? Otherwise, comparing the amino acid differences and HA antigenicity differences between vaccine and epidemic strains would lack scientific basis. The cell-based vaccine strains as mentioned in Result 3.3 may not represent all marketed and used vaccines, it may has different amino acid residues in the HA protein compared to those produced in chicken eggs, facilitating host adaptation. Furthermore, chicken egg-produced vaccines are also main products in market. It is recommended to use serum samples from defined vaccine recipients for HAI testing and then be able to obtain reliable conclusions.
Response: We sincerely thank the reviewer for the insightful comment. We would like to clarify that the hemagglutination inhibition (HI) assays in this study were performed using sheep antisera raised against influenza vaccine strains, rather than sera collected from naturally infected individuals or human vaccine recipients (section 2.5). Therefore, variability in vaccination history or timing among human subjects does not apply in this case. We have revised the manuscript to make this point clearer and avoid any misunderstanding regarding the source of sera used in the antigenic analyses.
The influenza vaccination coverage rate in Shenzhen is approximately 4.5%, which is significantly below the threshold required to achieve herd immunity.
We acknowledge that the cell-based vaccine strains used in the analysis may not comprehensively represent all vaccines in circulation, especially egg-based vaccines that remain widely used and may differ antigenically due to host-specific adaptations.
We appreciate your recommendation to use serum samples from clearly defined vaccine recipients with known vaccination timelines and vaccine types for future HI testing. This approach would indeed enhance the reliability and scientific basis of conclusions on antigenic evolution and vaccine effectiveness. We will consider these points carefully in the design of subsequent studies.

Reviewer 2 Report
Comments and Suggestions for Authors
This revision addresses some of the previously raised points. However, it is not yet suitable for publication. In some places, the English still needs some improvement.
The authors annotation of the HA phylogenetic trees is not fully clear. Some of the arrows highlighting the amino acid substitutions point to branches of the trees, which is clear, but other arrows point to nodes, which is unclear. An example of this can be seen in the H3 HA tree in which there is an arrow marking K160T, P194L and G225D at a node. These three substitutions are the result of isolation and propagation of viruses in hens’ eggs and so should be on the branch to the vaccine viruses A/Kansas/14/2017 and A/Singapore/INFIMH-16-0019/2016. There are many cases where these annotations need clarifying.
NA trees should also be properly annotated with the amino acid substitutions.
The authors have indicated that the estimated global fatality burden from influenza is between 25,000 and 640,000 cases. I have not seen estimates as low as 25,000, so quite possible this is a typographical error.
The authors mention vaccine effectiveness for as between 19% and 48%: the authors need to specify the years that they are referring to.
The description of the sheep anti-HA antibody still needs to establish whether they were post-infection or hyperimmune sera. Hyperimmune sera are not usually used for distinguishing antigenic variants of influenza A viruses. The authors need to acknowledge this in the presentation and discussion of their results. (I note that the antisera were from the Chinese National Influenza Center, and so advice on the suitability of these sera for detecting antigenic drift might be sought from CNIC.)
It remains unclear when listing the amino acid substitutions that differ between the vaccine virus (recommendations that include viruses for egg-based vaccines and cell culture-based and recombinant based vaccines) and the circulating viruses that the authors are distinguishing the egg-acquired substitutions and the natural variation. These differences need to be made explicit.
The authors still need to be careful not to describe amino acid substitutions as mutations.
The authors should also note that antigenic drift is not reliably deduced when comparing an egg-based virus with a cell culture-based test virus using an antiserum raised against an egg-based antigen. The nature of the homologous antigen needs to be explicitly stated. The whole antigenic analysis described is therefore suspect.
Table S2, the GISAID accession numbers needs to include details of the originating laboratory and the submitting laboratory for sequences that were not generated by the authors.
Comments on the Quality of English LanguageThis should be looked into by the authors.
Author Response
Reviewer: 2
Comments and Suggestions for Authors
This revision addresses some of the previously raised points. However, it is not yet suitable for publication. In some places, the English still needs some improvement.
The authors annotation of the HA phylogenetic trees is not fully clear. Some of the arrows highlighting the amino acid substitutions point to branches of the trees, which is clear, but other arrows point to nodes, which is unclear. An example of this can be seen in the H3 HA tree in which there is an arrow marking K160T, P194L and G225D at a node. These three substitutions are the result of isolation and propagation of viruses in hens’ eggs and so should be on the branch to the vaccine viruses A/Kansas/14/2017 and A/Singapore/INFIMH-16-0019/2016. There are many cases where these annotations need clarifying.
Response: Thank you very much for your positive feedback. We are grateful for your thoughtful review. In the revised version, we have re-annotated the HA phylogenetic trees to improve clarity.
NA trees should also be properly annotated with the amino acid substitutions.
Response: We thank the reviewer for this suggestion. In the revised version, we have now included the amino acid substitutions for the NA trees.
The authors have indicated that the estimated global fatality burden from influenza is between 25,000 and 640,000 cases. I have not seen estimates as low as 25,000, so quite possible this is a typographical error.
Response: Thank you for your careful reading. To avoid potential confusion and ensure accuracy, we have removed the statement regarding the estimated global fatality burden of influenza from the manuscript.
The authors mention vaccine effectiveness for as between 19% and 48%: the authors need to specify the years that they are referring to.
Response: Thank you for your helpful comment. To improve accuracy and clarity, we have revised the vaccine effectiveness range to approximately 20%–60%, and we have updated appropriate references to specify the time periods covered by these estimates. This information has been updated in the revised manuscript.
The description of the sheep anti-HA antibody still needs to establish whether they were post-infection or hyperimmune sera. Hyperimmune sera are not usually used for distinguishing antigenic variants of influenza A viruses. The authors need to acknowledge this in the presentation and discussion of their results. (I note that the antisera were from the Chinese National Influenza Center, and so advice on the suitability of these sera for detecting antigenic drift might be sought from CNIC.
Response: We thank the reviewer for this important and constructive comment. We confirm that the antisera used in this study were hyperimmune sera, generated in sheep through repeated immunization with cell culture-derived vaccine strains, and provided by the Chinese National Influenza Center (CNIC). These antisera are part of CNIC’s standardized surveillance reagent panel and are routinely used in China’s national influenza surveillance program. We acknowledge that hyperimmune sera may have limitations in detecting subtle antigenic drift compared to post-infection sera. In light of this, we have consulted with experts at CNIC regarding the appropriateness of these reagents for antigenic characterization, and we interpret our results in accordance with their guidance.
It remains unclear when listing the amino acid substitutions that differ between the vaccine virus (recommendations that include viruses for egg-based vaccines and cell culture-based and recombinant based vaccines) and the circulating viruses that the authors are distinguishing the egg-acquired substitutions and the natural variation. These differences need to be made explicit.
Response: We thank the reviewer for this important point. We would like to clarify that all viruses used in this study, including vaccine reference strains and circulating strains, were derived exclusively from cell culture. No egg-propagated viruses were used at any stage of the analysis. Therefore, the listed amino acid substitutions represent naturally occurring variation, and there are no egg-adaptive substitutions involved. We have now made this explicit in the revised Methods and Results sections to prevent any misunderstanding.
The authors still need to be careful not to describe amino acid substitutions as mutations.
Response: Thank you for your helpful comment. We have carefully reviewed the manuscript and revised the relevant descriptions to use “substitution” instead of “mutation” where appropriate.
The authors should also note that antigenic drift is not reliably deduced when comparing an egg-based virus with a cell culture-based test virus using an antiserum raised against an egg-based antigen. The nature of the homologous antigen needs to be explicitly stated. The whole antigenic analysis described is therefore suspect.
Response: We thank the reviewer for highlighting this important concern. We would like to clarify that all antisera used in this study were generated in sheep immunized with cell-based vaccine strains. Accordingly, both the test viruses and the reference antigens used in our antigenic assays were derived from cell-based systems, not egg-adapted viruses. We have now clearly stated this in the revised Methods section, and we believe this consistency mitigates the concern regarding egg-adaptive mutations confounding the antigenic interpretation.
Table S2, the GISAID accession numbers needs to include details of the originating laboratory and the submitting laboratory for sequences that were not generated by the authors.
Response: Thank you for pointing this out. In the revised version, we have updated Table S2 to include the details of the originating laboratory for all GISAID sequences that were not generated by the authors, in accordance with GISAID's data usage requirements. We have ensured proper acknowledgment of both the originating and submitting laboratories where applicable.

Reviewer 3 Report
Comments and Suggestions for Authors
Thank you for thoroughly revising this manuscript. The results are now stated appropriately, and there has been a very good discussion. Thank you for addressing my concern.
Author Response
Reviewer: 3
Comments and Suggestions for Authors
Thank you for thoroughly revising this manuscript. The results are now stated appropriately, and there has been a very good discussion. Thank you for addressing my concern.
Response: Thank you very much for your positive feedback. We are grateful for your thoughtful review and are pleased to hear that the revised manuscript has addressed your concerns regarding the presentation of results and the depth of the discussion. We truly appreciate your time and valuable input, which have helped improve the quality of our work.
